# Inducción de árboles de decisión mediante colonia de hormigas para el problema de *label ranking*

**Juan C. Alfaro**
Departamento de Sistemas Informáticos
Universidad de Castilla-La Mancha
Albacete, España, 02071
JuanCarlos.Alfaro@uclm.es

**Juan A. Aledo**
Departamento de Matemáticas
Universidad de Castilla-La Mancha
Albacete, España, 02071
JuanAngel.Aledo@uclm.es

**José A. Gámez**
Departamento de Sistemas Informáticos
Universidad de Castilla-La Mancha
Albacete, España, 02071
Jose.Gamez@uclm.es

## Abstract

El problema de *label ranking* es una tarea de clasificación supervisada no estándar cuyo objetivo es predecir un orden total de las etiquetas de la variable clase para una instancia de entrada. Además, las instancias del conjunto de datos de entrenamiento también están etiquetadas con este tipo de órdenes. Entre los enfoques existentes, el algoritmo basado en instancias ha demostrado ser uno de los más competitivos. Sin embargo, solo es superado por métodos de *ensemble*, que, al no ser fácilmente interpretables, pueden no ser adecuados en ámbitos donde la transparencia sea fundamental. Para abordar esta limitación, en este trabajo se propone el diseño de un método de inducción de árboles de decisión basado en colonia de hormigas, con el objetivo de: (1) mejorar la tasa de acierto respecto a los enfoques basados en instancias, y (2) generar modelos más interpretables que los métodos de *ensemble*. A diferencia de los métodos heurísticos voraces tradicionalmente utilizados en este problema, mediante la metaheurística de colonia de hormigas se busca explorar un espacio de búsqueda más amplio. La propuesta será evaluada experimentalmente en conjuntos de datos estándar, comparando su desempeño con los métodos existentes en términos de tasa de acierto, coste computacional y complejidad del modelo.

## 1. Motivación e hipótesis de trabajo

En una amplia variedad de aplicaciones de aprendizaje automático, especialmente aquellas que requieren intervención humana, como la evaluación de tratamientos médicos, encuestas de opinión, competiciones deportivas, entre otras, los datos disponibles suelen ser cualitativos en lugar de cuantitativos. Un ejemplo reciente en el que se han realizado contribuciones significativas es en el proceso de *fine-tuning* de modelos extensos de lenguaje [14]. En este caso, dado un *prompt*, el modelo genera varias respuestas que un humano ordena de peor a mejor, sin necesidad de asignar puntuaciones numéricas a las salidas. Esta ordenación, que representa información puramente cualitativa, se utiliza para aprender una función de recompensa que permite ajustar el modelo y mejorar su rendimiento.

El ejemplo anterior ilustra la tarea de aprendizaje en el problema de *label ranking* [21]. En este escenario, el objetivo es aprender un modelo de preferencia, denominado *label ranker*, que prediga un orden total de las etiquetas disponibles (respuestas) dada una instancia de entrada (*prompt*), con el

fin de que esta predicción coincida con el *ground truth* correspondiente. Este problema forma parte del aprendizaje de preferencias [13] y se ha aplicado a diversos escenarios, como la clasificación multietiqueta [11], la selección de algoritmos [12] y la estimación de profundidad monocular [15], entre otros.

A lo largo de los años, se han propuesto diversos algoritmos para abordar el problema de *label ranking*, incluyendo enfoques basados en instancias [6, 7], inducción de árboles de decisión [7, 20, 19], *ensembles* [1, 2, 9, 10, 19, 23], redes neuronales [17] y modelos basados en mixturas [18, 22], entre otros.

Sin embargo, los enfoques basados en instancias han demostrado un rendimiento superior en términos de tasa de acierto a todas las técnicas basadas en modelo, con la excepción de los *ensembles*. A pesar de su mayor rendimiento, estos últimos presentan la desventaja de perder interpretabilidad debido a la agregación de múltiples modelos, lo que puede restringir su uso en contextos donde, además de obtener buenos resultados, es fundamental garantizar la transparencia en la toma de decisiones, como en el ámbito médico.

Motivados por la falta de modelos individuales capaces de superar a los enfoques basados en instancias en el problema de *label ranking*, este trabajo se enfoca en el diseño de algoritmos basados en la inducción de árboles de decisión mediante colonias de hormigas [3, 4, 16]. La novedad de esta investigación radica en la aplicación de metaheurísticas para explorar un espacio de soluciones más amplio y diverso, con el objetivo de encontrar modelos globalmente óptimos, ya que los métodos disponibles en la literatura se basan principalmente en procesos heurísticos voraces estándar y suelen quedar restringidos a óptimos locales.

Como hipótesis de trabajo, planteamos que el uso de metaheurísticas, específicamente colonias de hormigas, permitirá superar las limitaciones de los métodos actuales, generando *label rankers* con un rendimiento superior y manteniendo la interpretabilidad inherente de los árboles de decisión. Si bien el enfoque propuesto puede implicar un mayor tiempo de entrenamiento en comparación con las técnicas existentes, su potencial para generar modelos con mejores resultados sin sacrificar la interpretabilidad lo convierte en una alternativa prometedora.

## 2. Objetivos

El objetivo principal de esta propuesta es el estudio, diseño, implementación y validación de un algoritmo de inducción de árboles de decisión basado en colonias de hormigas para abordar el problema de *label ranking*, con el propósito de superar el rendimiento de los algoritmos basados en instancias en términos de precisión predictiva, al mismo tiempo que se garantiza la interpretabilidad de los modelos obtenidos para su uso en la toma de decisiones. Para alcanzar el objetivo general, se plantean los siguientes objetivos específicos:

1. Diseño de un grafo de construcción adaptado al problema de *label ranking*. El grafo de construcción es una estructura que guía a las hormigas en la exploración del espacio de soluciones. Habitualmente, los nodos son atributos (nominales o continuos) y las aristas representan las condiciones para construir árboles de decisión. Para atributos nominales, cada arista representa una condición específica sobre los valores del atributo. En el caso de atributos continuos, existen dos alternativas principales: (1) utilizar el procedimiento estándar, donde cada valor diferente de la variable numérica representa una posible arista, o (2) aplicar técnicas de discretización supervisada basadas en el principio de *minimum description length*. Nuestro objetivo aquí es implementar técnicas de discretización supervisada en el grafo de construcción, adaptadas específicamente al problema de *label ranking*, utilizando métodos como los propuestos en [8].

2. Incorporación de información heurística específica para el problema de *label ranking*. La información heurística sirve para guiar a las hormigas en la selección de atributos y condiciones durante la construcción del árbol de decisión, priorizando aquellos que maximicen la capacidad del modelo para predecir correctamente la variable objetivo correspondiente. En problemas de clasificación supervisada, se utiliza comúnmente la ganancia de información como heurística. Sin embargo, en *label ranking*, se han propuesto diversos criterios, como el parámetro de dispersión del modelo de probabilidad *Mallows* [7] o la distancia de *Kendall* [19], entre otros. El objetivo aquí es analizar estos criterios y seleccionar el más

apropiado en términos de capacidad predictiva y eficiencia computacional, asegurando que la heurística elegida sea efectiva para guiar la construcción de árboles de decisión en el problema de *label ranking*.

3. Definición de la matriz de feromonas. La matriz de feromonas es una estructura clave que almacena la información sobre la calidad de las decisiones tomadas por las hormigas durante la construcción de los árboles, permitiendo reforzar las decisiones más prometedoras en iteraciones posteriores. Estas decisiones se limitan a la selección de una condición específica de un atributo en un nivel determinado del árbol, por lo que el objetivo aquí es estudiar cuál es la estructura de datos más eficiente para representar esta información, considerando factores como la escalabilidad del algoritmo y la complejidad computacional.

4. Desarrollo de un mecanismo de construcción de árboles de decisión basado en colonias de hormigas. En este objetivo, implementamos un procedimiento en el que las hormigas seleccionan atributos de manera probabilística, combinando la información heurística y la matriz de feromonas. De esta manera, las hormigas exploran el espacio de soluciones de forma guiada, priorizando atributos y condiciones que han demostrado ser efectivos en iteraciones anteriores (feromonas) y que son prometedores según criterios locales (heurística), manteniendo un equilibrio entre exploración y explotación del espacio de soluciones.

5. Implementación de un mecanismo de actualización de la matriz de feromonas. El objetivo es actualizar la matriz de feromonas basándonos en la calidad de los árboles construidos, utilizando métricas de evaluación específicas para *label ranking*. Habitualmente, se utiliza el coeficiente de correlación de *Kendall* cuando se desea una métrica equivalente a la tasa de acierto, o la distancia de *Kendall* en caso de medir el error.

6. Diseño de un procedimiento de poda para mejorar la generalización e interpretabilidad del modelo. Una de las técnicas más utilizadas en la literatura para la poda es el criterio *minimal cost-complexity* [5], ya que proporciona un equilibrio entre la complejidad del árbol y su error de predicción. El objetivo es adaptar este procedimiento al problema de *label ranking*, utilizando métricas como la distancia de *Kendall* para evaluar el error en la predicción.

## 3. Metodología

Este trabajo aborda un desarrollo metodológico, por lo que se empleará el método científico para abordar su componente investigadora. En primer lugar, se llevará a cabo una revisión exhaustiva de los métodos existentes para el problema de *label ranking*, con especial énfasis en los métodos de inducción de árboles de decisión y el uso de colonias de hormigas en otras tareas de aprendizaje supervisado. A continuación, se diseñará un algoritmo de inducción de árboles de decisión basado en colonias de hormigas, incorporando modificaciones específicas para abordar las particularidades del problema de *label ranking*. Finalmente, se validará el rendimiento del algoritmo propuesto mediante conjuntos de datos consensuados por la comunidad científica y técnicas de validación estándar. Además, se analizará la complejidad de los árboles generados mediante métricas como la profundidad, el número de nodos internos y nodos hoja. Este análisis permitirá evaluar la interpretabilidad de los modelos y su relación con el rendimiento predictivo, asegurando que el algoritmo no solo sea preciso, sino también comprensible y aplicable en contextos donde la transparencia es fundamental.

## Agradecimientos

Este trabajo está parcialmente financiado por los siguientes proyectos: SBPLY/21/180225/000062 (Junta de Comunidades de Castilla-La Mancha y FEDER, Una manera de hacer Europa), PID2022-139293NB-C32 (MICIU/AEI/10.13039/501100011033 y FEDER, UE), y 2022-GRIN-34437 (Universidad de Castilla-La Mancha y FEDER, Una manera de hacer Europa).

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
