# OpenReview forum: "Inducción de árboles de decisión mediante colonia de hormigas para el problema de label ranking"
_MAEB/2025/Projects_Track — MAEB 2025 Proyectos_

### Official Review · Reviewer_qeQ9 · 2025-03-11
**Propuesta prometedora, a falta de detalles**

**Rating:** 4
**Confidence:** 3

**Review:**

La descripción del proyecto se enfoca en el problema de "label ranking": es decir, en la tarea de clasificación supervisada que busca predecir el orden de las etiquetas de una variable de clase.

La motivación para el estudio del problema es sólido, aunque recomendaría extender la descripción (y posteriormente, evaluación) del estado del arte (incluyendo más detalles sobre cómo operan los algoritmos de instancia y los de inducción de árboles de decisión, y sus limitaciones).

El objetivo del proyecto es proponer un nuevo método para el problema de "label ranking", basado en árboles de decisión y colonias de hormigas, buscando:

- Mejorar la precisión de los algoritmos basados en instancias.
- Mejorar la interpretabilidad de los métodos de ensemble.

El argumento de los autores para utilizar la metaheurística de colonias de hormigas es que éstos permitirán explorar un espacio de búsqueda más amplio. Es decir, los autores exponen su hipótesis principal de que la metaheurística de colonias de hormigas permitirá "explorar un espacio de soluciones más amplio y diverso, con el objetivo de encontrar modelos globalmente óptimos, ya que los métodos disponibles en la literatura (...) suelen quedar restringidos a óptimos locales."

El proyecto, por tanto, consiste en diseñar y experimentar con un algoritmo que valide tal hipótesis.

En principio, la propuesta es razonable, si bien queda supeditada a la ejecución de los experimentos descritos (sin garantías teóricas).

Los autores planean diseñar el modelo en base a un grafo de construcción adaptado al problema de label ranking, con su correspondiente definición de la matriz de feromonas y la incorporación de información del problema en la optimización heurística. En este punto, sería recomendable proveer más detalles sobre los criterios de heurística (Mallows Vs Kendall) y sus implicaciones prácticas.

Finalmente, los autores proponen evaluar el rendimiento del método en conjuntos de datos estándar, comparándolo con otros métodos en términos de precisión, costo computacional y complejidad del modelo. Sin embargo, hecho en falta detalles sobre cómo se va a evaluar la interpretabilidad de los modelos, siendo éste uno de los objetivos primordiales del trabajo.

En conclusión, el proyecto presenta una propuesta prometedora para abordar el problema de "label ranking", pero la falta de detalles técnicos y la necesidad de detalles más rigurosos sobre las métricas de diseño y evaluación (sobre todo de la interpretabilidad de los métodos) son aspectos que deben abordarse para garantizar su éxito.

---

### Official Review · Reviewer_fSMq · 2025-03-19
**Los autores proponen el diseño de un algoritmo basado en árboles de decisión y colonia de hormigas para la solución del problema de 'label ranking'. Fundamentan el uso de la metaurística para mantener un sustrato de interpretatividad. El problema es interesante y se agradece la descripción exhaustiva de los objetivos. Sin embargo, dentro  del proyecto descrito deben aclararse o justificarse ciertos aspectos importantes que se presentan en los comentarios de la revisión.**

**Rating:** 3
**Confidence:** 5

**Review:**

Los autores proponen el diseño de un algoritmo basado en árboles de decisión y colonia de hormigas para la solución del problema de 'label ranking'. Fundamentan el uso de la metaurística para mantener un sustrato de interpretatividad. El problema es interesante y se agradece la descripción exhaustiva de los objetivos.

Comentarios

En general dado el planteamiento, los autores pretenden realizar una hibridación (soft o hard) de dos métodos específicos como son los árboles de decisión y las colonias de hormigas. Hay casos en la literatura al respecto, como por ejemplo, DOI: 10.1007/978-3-319-93752-6_3.

Un elemento importante a la hora de trabajar con metaurísticas es elegir el método adecuado del problema por lo que sería importante justificar porque se decantó el uso de la colonia de hormigas por encima de otros métodos. Por otra parte, es importante analizar los parámetros propios del algoritmo que pueden incidir en el rendimiento correspondiente.

Además, al ser la colonia hormiga un método estocástico debe incluirse algún análisis estadístico acerca de los resultados presentados, así como, el conjunto de instancias sobre el cual se está realizando o se realizará el proceso de evaluación.

---

### Decision · Program_Chairs · 2025-03-20

Accept